# Adsorption of Methylene Blue and Tetracycline by Zeolites Immobilized on a PBAT Electrospun Membrane

**DOI:** 10.3390/molecules28010081

**Published:** 2022-12-22

**Authors:** David Picón, Alicia Vergara-Rubio, Santiago Estevez-Areco, Silvina Cerveny, Silvia Goyanes

**Affiliations:** 1Laboratorio de Polímeros y Materiales Compuestos (LP&MC), Departamento de Física, Facultad de Ciencias Exactas y Naturales, Universidad de Buenos Aires, Ciudad Universitaria (C1428EGA), Ciudad Autónoma de Buenos Aires B1428, Argentina; 2Instituto de Física de Buenos Aires (IFIBA), CONICET—Universidad de Buenos Aires, Ciudad Universitaria (C1428EGA), Ciudad Autónoma de Buenos Aires B1428, Argentina; 3IIIA-UNSAM-CONICET, Instituto de Investigación e Ingeniería Ambiental, Escuela de Hábitat y Sostenibilidad, San Martín B1650, Argentina; 4Instituto de Física de Materiales Tandil (UNCPBA) and CIFICEN (UNCPBA-CICPBA-CONICET), Pinto 399 (B7000GHG), Tandil B7000, Argentina; 5Centro de Física de Materiales (CSIC, UPV/EHU)-Materials Physics Center (MPC), Paseo Manuel de Lardizabal 5, 20018 San Sebastián, Spain; 6Donostia International Physics Center (DIPC), 20018 San Sebastián, Spain

**Keywords:** clinoptilolite zeolite, electrospinning, emerging contaminant removal, membrane modification, PBAT

## Abstract

The detection of emerging contaminants in bodies of water has steadily increased in recent years, becoming a severe problem threatening human and ecosystem health. Developing new materials with adsorption properties to remove these pollutants represents an important step toward a potential solution. In this paper, a polybutylene adipate terephthalate (PBAT) nanofibrous membrane incorporating clinoptilolite zeolite was developed and its excellent performance in removing tetracycline (TC) and methylene blue (MB) from water was demonstrated. The composite membrane was prepared in two steps: firstly, a homogeneous dispersion of clinoptilolite (1 wt% respect to polymer) in a PBAT solution (12.6 wt%) was electrospun; secondly, the electrospun membrane was subjected to an acid treatment that improved its wettability through the protonation of the surface silanol groups of clinoptilolite. The resulting membrane was hydrophilic and showed higher adsorption for TC (800 mg/g) and MB (100 mg/g), using a low dose (90 mg/L) powdered zeolite. The maximum removal capacity was obtained at neutral pH, being the cation exchange reaction the main adsorption mechanism. Pseudo-second-order kinetics and Henry’s law agree well with the proposed chemisorption and the high affinity of TC and MB for the adsorbent. The material can be reused after the removal process without generating additional contamination, although losing some effectivity.

## 1. Introduction

Water pollution is among the world’s most serious issues, affecting human health and ecosystem life. Approximately 771 million people (1 in 10) globally do not have access to safe water. This emergency carries a significant problem: drinking contaminated water makes people sick [1]. It is also a problem producing fresh food such as fresh fruit and vegetables. Unsafe water can carry harmful microorganisms from animal or human feces, and this greatly influence the risk of crop contamination [2]. Because of these problems, several countries have adopted legislation and policies to improve water quality and avoid contamination. In parallel, numerous scientific and technological advances have been made globally to mitigate this problem [1,3].

High concentrations of specific heavy metals, either because of natural or anthropogenic causes, have historically been the primary concern regarding water contamination. However, detecting large amounts of the so-called emerging pollutants in water has emerged in recent years as another serious issue. This concept includes various synthetic compounds widely used in contemporary societies, including cosmetics, insecticides, antibiotics, and dyes [4]. The emerging pollutants present a fast growth rate, and there are a significantly fewer regulations and investigations focused on their detection and potential treatments compared with the ‘traditional’ pollutants [5,6].

From all the possible emerging contaminants, in this work, the focus here is on dyes and antibiotics. Dyes are primarily used in the production of consumer products, including paints, textiles, printing inks, paper, and plastics. Synthetic dyes are used in the industry because they bond better to fabric, providing an intense color that lasts after washing and exposure [7]. Further, synthetic dyes can be produced at a large scale, requiring the supplies to be inserted in the production chain of large industries. Direct discharge of untreated effluents from textile industries leads to water pollution. In particular, methylene blue (MB) dye is a heterocyclic compound (C_16_H_18_N_3_SCl) typically used in many textile products as a coloring agent. MB is a toxic, environmentally persistent, non-biodegradable cationic dye. It is stable to light, heat and oxidation, and due to these properties, MB is used as a model for other hazardous organic contaminants in water [8,9]. Thus, MB in water supplies damages the environment and human health, causing several diseases [10].

Antibiotics are used in human and veterinary medicine to treat and prevent infectious diseases. Veterinary antibiotics increase feed utilization efficiency, promote animal growth, and decrease mortality and morbidity in livestock and poultry industries [11]. However, antibiotics are poorly metabolized by organisms: generally, 50–80% of ingested antibiotics are excreted and discharged into the environment, mainly into water bodies [12]. In recent years, different studies have warned of the presence of antibiotics in several countries’ rivers, streams, lakes, and groundwater [13]. Tetracycline antibiotics are broad-spectrum antibiotics widely used in human and animal health, ranking second worldwide in use and production. Chlortetracycline, oxytetracycline, and tetracycline (TC) are the most common antibiotics of this kind [14]. TC residues released into the environment contaminate soils and waters, and their presence could increase bacterial resistance and the risk of toxicity of aquatic species or human beings if ingested [15].

Nowadays, different technologies allow the removal of contaminants from water, including coagulation/flocculation, photo-degradation, filtration, sedimentation, adsorption, and ion exchange [16,17]. Adsorption is recognized as a low-cost and highly efficient process that can target several contaminants at once. Recently, there has been increasing interest in removing emerging pollutants from water using naturally occurring adsorbents such as clay minerals [18,19], biomass [20,21] and zeolites [22,23]. In particular, natural zeolites, such as clinoptilolite, present an outstanding performance in removing contaminants from water. The adsorption capacity of zeolites is based on their high cation exchange capacity and cage-type structure [24,25]. However, due to the physical granular powder form of zeolites, this kind of adsorbent needs to be removed after the remediation process by a filtration step. Alternatively, as proposed in this work, zeolites could be dispersed in physical support, and therefore, they can be easily removed after the adsorption process.

Electrospinning constitutes a simple and effective technology for fabricating sub-micron sized fibers from different polymers. This technique is industrially scalable and allows incorporating and immobilizing adsorbents in a single state; for example, the addition of iron nanoparticles, carbonaceous nanoparticles, L-cysteine, and polyphenols-rich extracts to electrospun fibers has been studied recently [26,27,28,29,30]. Polybutylene adipate terephthalate (PBAT) is a non-toxic, soil-biodegradable and water-insoluble polymer from which nanofibers can be electrospun [31,32]. PBAT-based nanofibers have good mechanical properties, such as high elasticity, so they could adequately support zeolites. PBAT-based electrospun fibers have been used for different applications: in designing bactericidal wound dressings from PBAT/PLA blends with immobilized zinc phthalocyanine [33], and PBAT fibers loaded with gentamicin [34]. However, PBAT fibers are hydrophobic, so their poor interaction with water would restrict their use in pollutant removal.

PBAT and zeolites have been combined to produce composites with different applications: PBAT/zeolite 13X film was developed to increase the shelf life of bananas [35], and PBAT/zeolite Y films capable of controlling the release of rhynchophorol pheromone [36]. PBAT/synthetic zeolite films were also used to develop biodegradable active packaging with ethylene scavenge properties [37]. In the case of water remediation, PBAT nanofibers have been used to adsorb endocrine disruptor compounds (estrone and 17β-estradiol) or associated with silver nanoparticles to remove bacteria [38,39]. Although PBAT nanofibers have been combined with different adsorbents to remove pollutants from water, there are no previous reports using electrospun PBAT membranes filled with zeolites for water remediation applications. This is because, as explained above, PBAT is hydrophobic and, therefore, seems unsuitable for this application.

Recently, in our previous works, it has been detected that acid treatments can change the hydrophobic character of electrospun membranes to hydrophilic [40]. Additionally, it has been reported that acid treatments of zeolites with hydrochloric acid (HCl) increase their interaction with pollutants such as lead and cadmium [41]. Based on this, an acid treatment on a PBAT membrane containing zeolites could be an effective approach to increase its interaction with water and maximize pollutant removal. Even though zeolites are often used in combination with other polymers [42,43], there are no precedents in which they are combined with PBAT nanofibers as a strategy to ensure optimal zeolite dispersion for use in removing water contaminants. Additionally, acid treatments to improve the interaction with water of PBAT-zeolites have not been reported either.

This work proposes a new electrospun PBAT membrane to confine clinoptilolite zeolites in a self-supporting and macroscopic material. In this way, zeolites are both adequately distributed (no aggregation) and immobilized between PBAT nanofibers, maximizing the surface/volume ratio available for adsorption. Because of the hydrophobicity of PBAT, the resulting material was further exposed to an acid treatment to activate the zeolites and increase the contact between water and the polymer. The performance of the PBAT nanofibers filled with zeolites in removing tetracycline (TC) and methylene blue (MB) from the water was evaluated. In addition, the influence of pH on adsorption and removal efficiency in reuse was studied. Finally, the main adsorption mechanism of MB and TC in this novel material was discussed.

## 2. Results and Discussions

### 2.1. Morphology of Zeolites and PBAT Membranes

Figure 1a shows a representative SEM image of zeolites. They are formed by spherical structures with diameters of 0.5 to 1 µm. On the other hand, the morphology of PBAT-Z membranes consists of the typical structure of intertwined nanofibers, among which the zeolites are well distributed (Figure 1b). PBAT nanofibers are cylindrical with diameters of 100–250 nm, while zeolites have a spherical shape of 1–8 µm trapped between the nanofibers (Figure 1c).

PBAT and PBAT-Z membranes dyed with rhodamine B were examined by fluorescence microscopy. As expected, the PBAT membrane did not present fluorescence, confirming that the polymer cannot adsorb rhodamine B (data not shown). On the other hand, PBAT-Z membrane images show fluorescent circles of diameters between 1 and 10 µm, which correlate to the spheres observed by SEM microscopy (Figure 1d). This image confirms the presence of zeolite in the composite material, whereas PBAT fibers cannot be detected since they cannot adsorb rhodamine B.

Furthermore, the presence of zeolite among the PBAT fibers was confirmed by EDS analysis, as reported in Figure 1e. The relative intensity of the signals in the EDS spectrum should not be considered as the material of the cue, and the carbon ribbon influences it.

### 2.2. Physicochemical Properties of Acid-Treated Membranes

This section describes the effect of acid treatment on the wettability of PBAT-Z membranes. Contact angle, FTIR and TGA studies were performed to elucidate the mechanisms responsible for wettability modification.

Figure 2 shows the contact angle of PBAT-Z membranes treated with different concentrations of HCl. As a control, the contact angle was measured on PBAT membranes without zeolites. The untreated PBAT-Z membrane has a contact angle of ~108°, correspondingly with a hydrophobic material. For the treated samples, the contact angle diminishes with increasing acid concentration, indicating that the membrane becomes more hydrophilic. In particular, for PBAT-Z-6M, the contact angle was ~16°, whereas the PBAT membrane without zeolites remains hydrophobic even after the acid treatment with a concentration of 6 M. This result implies that the acid treatment only modifies the zeolites, whereas the polymer remains hydrophobic.

The FTIR spectra of PBAT-Z membranes treated with different concentrations of HCl are shown in the 2000–600 cm^−1^ (Figure 3a) and 4000–2400 cm^−1^ (Figure 3b) regions. The strong band observed at 2946 cm^−1^ corresponds to C-H stretching vibration of aliphatic and aromatic portions, and the weak and broad band centered at ~3450 cm^−1^ was attributed to –OH stretching vibrations arising from surface silanol groups (≡Si-OH) from zeolite structure, which are acid Brønsted sites [44,45]. In the 2000-600 cm^−1^ region, the characteristic bands of PBAT were observed: C=O stretching (1710 cm^−1^), C-H bending (1455 and 1408 cm^−1^), C-O stretching (1265 and 1100 cm^−1^) and C-H_2_ bending (724 cm^−1^) [46]. It should be mentioned that the acid treatment does not produce any observable changes in the neat PBAT FTIR spectrum (Appendix A), in agreement with the contact angle result for PBAT membranes (Figure 2).

FTIR spectra of PBAT-Z membranes after acid treatment did not show differences in the 2000–600 cm^−1^ region compared with untreated membranes (Figure 3a). This result indicates no interactions, detectable by FTIR, between zeolite and PBTA in the region of the PBAT fingerprints, even after different acid treatments. The thermal degradation of PBAT-Z and PBAT-Z-6M also showed no differences, confirming that the polymer does not suffer chemical transformations with the addition of zeolite or acid treatment (see TGA analyses shown in the Appendix A).

On the other hand, in the 4000–2400 cm^−1^ region (Figure 3b), a significant increase in the intensity of the –OH band was observed in the PBAT-Z sample treated with 6M HCl aqueous solution, which correlates with the observed change in the wettability (Figure 2). The difference in the intensity of –OH band could be attributed to the formation of Si-OH_2_^+^ groups in the zeolites according to the following reactions [47,48]:SiOH + H^+^ → SiOH_2_^+^(1)

Silanol group can also be deprotonated under basic conditions:SiOH + OH^−^ → SiO^−^ + H_2_O(2)

Further, acidic treatment (6M) on the zeolites (without membrane) was performed. Figure 3c shows the comparison of FTIR spectra between treated and untreated zeolites. In such a case, the most intense bands are related to the presence of zeolitic water (3000–3800 cm^−1^). In addition, an asymmetric internal T-O stretching vibration (T = Si or Al) (1029 cm^−1^) can also be seen [41,49]. After acid treatment, the T-O stretching band significantly shifts to higher wavenumbers. Given that the position of this band depends on the Si/Al ratio, it is concluded that the acid treatment produces zeolites with a lower Al/Si ratio [50,51].

Protonation of zeolites improves the contact of the material with the aqueous solution because the modified zeolites have a more significant interaction with water. The acid treatment on zeolites also promotes the formation of Brønsted sites through a dealuminization reaction [52]. Thus, the increase in Brønsted sites in the zeolite surface led to a higher interaction with water molecules. As more silanol groups are formed in the zeolite surface, the number of active adsorption sites increases. Additionally, it has been reported that after acid treatment, all the original cations (i.e., K^+^, Na^+^, and Mg^2+^) in the clinoptilolite framework are replaced by hydronium ions [52], which facilitates the cation exchange.

Considering the contact angle and FTIR results, acid treatment with 6M HCl solution produces the most significant changes in membrane water wettability without altering the polymer structure. Therefore, the PBAT-Z-6M membrane was used in the subsequent experiments.

### 2.3. Adsorption Studies

#### 2.3.1. Performance of Electrospun PBAT-Z Mats and Powdered Zeolite after Treatment with HCl 6 M on Adsorption of TC and MB

Table 1 compares the adsorption capacity of TC and MB by PBAT, PBAT-Z mats, and powdered zeolite before and after acid treatment (HCl-6 M). The adsorption capacity of both contaminants was significantly higher in PBAT-Z-6M than the powdered zeolite-6M at the same dose for both adsorbents (90 mg/L). This enhancement is due to the excellent dispersion of zeolite in the PBAT electrospun mat, improving the zeolite exposed area and, thus, increasing the removal capacity. Further, the adsorption capacity of the PBAT-Z membrane was increased by ~3.7 (for TC) and ~1.5 (for MB) fold after the acid treatment. This confirms that the higher hydrophilicity of the membrane allows a better interaction between the adsorbate and the adsorbent. The formation of Si-OH^2+^ groups in zeolites after its reaction with HCl improves the interaction with the adsorbate. Subsequently, the incorporation of zeolites in the electrospun nanofibers allows immobilizing of the adsorbent and makes it easier to remove it after its use.

It is noteworthy that PBAT and PBAT-6M mats adsorb TC and MB, although their *q* values are low compared with those obtained for PBAT-Z membranes. Thus, the polymeric matrix (not filled with zeolites) contributes approximately 3% to the total adsorption for TC and only 0.11 mg/g for MB. This result may be due to the interactions between the aromatic ring of PBAT and the unsaturated (poly)cyclic molecules of the adsorbates (π–π stacking) [53]. Similarly, it has been reported that π–π interactions play a crucial role in the adsorption mechanism of endocrine disruptors in water by PBAT electrospun microfibers [38].

On the other hand, the adsorption of TC is markedly higher than that of MB under the same conditions. The charge of the adsorbent can explain the different affinity for MB or TC at a given pH. For instance, at pH 6.50, MB predominates as a cation [9] and TC as an anion, whereas the zeolite surface is positively charged. Therefore, TC adsorption onto zeolite surface is favored due to electrostatic interactions. The effect of pH on the adsorption capacity will be further discussed in Section 2.3.4.

#### 2.3.2. Adsorption Isotherm

The adsorption isotherms were obtained by representing the adsorption capacity (*q*) against equilibrium concentration at 24 h. Figure 4 displays the adsorption isotherms of the PBAT-Z-6M membrane for TC (a) and MB (b), showing a linear behavior for both contaminants in the range studied. Henry’s model is the most straightforward adsorption model in which the amount of adsorbate is proportional to the concentration, according to Equation (3):(3)$$q_{e}=K_{H}C_{e}$$
where *q_e_* (mg/g) is the amount of adsorption at equilibrium, *K_H_* (L/g) is Henry’s constant, and *C_e_* (mg/L) is the equilibrium concentration of adsorbate in solution.

This model describes the adsorption usually at low concentrations, as those analyzed in this work. For the same reason, Langmuir’s or Freundlich’s models do not apply to the present case. In addition, Henry’s model has already been used in literature to describe liquid-phase adsorption. For example, the adsorption of heavy metals [54,55], antibiotics [56] and dyes [57] from water by the solid adsorbent.

Therefore, the experimental data were fitted with Henry’s model, and the results are shown in Table 2. It is observed that the isotherm of adsorption has a linear behavior for both adsorbates (R^2^ > 0.96), as expected from Henry’s law.

Isotherm adsorptions reveal a high affinity between the zeolites and TC, mainly at low concentrations, resulting in a maximum adsorption capacity of ~800 mg TC/g zeolite. This value is significantly higher compared with previous reports in the literature for the adsorption of TC. For example, it was reported an adsorption capacity of ~255 mg/g for zeolitic imidazolate framework nanoparticles/polyacrylonitrile nanofibers membrane (ZIF/PAN) [42], ~150 mg/g for zero-valent iron/natural zeolites [58] and ~200 mg/g for Fe-doped Synthetic zeolite 13X [59]. Additionally, for unmodified clinoptilolite and Fe_3_O_4_/Clinoptilolite powders, a maximum adsorption capacity of 20.17 and 180.9 mg/g, respectively, have been reported for TC removal [60]. Note that, except for ZIF/PAN nanofibers, these zeolite-based adsorbents are powders, requiring an additional filtration step to separate them after the removal process. In addition, harmful and expensive chemicals such as sodium borohydride are used to synthesize iron nanoparticles, and therefore, the experimental methodology requires multiple stages, increasing the final adsorbent’s cost.

On the other hand, the maximum adsorption capacity for MB was ~100 mg/g for the PBAT-Z-6M mat in the studied range. This value is similar to or better than those reported by other authors; for example, a maximum adsorption capacity of ~105 mg/g using synthetic ZSM-5 zeolite [61], ~27 mg/g using clinoptilolite-clay composite [19] and 7.7 mg/g of MB employing hydrothermally treated clinoptilolite [62].

The previous comparison with other adsorbents to remove MB and TC shows the importance of both a good dispersion and immobilization of the zeolite on PBAT nanofibers, avoiding zeolite agglomeration, improving its contact area and, therefore, its adsorption capacity, but also facilitating the removal of the adsorbent after its use.

#### 2.3.3. Adsorption Kinetics

The adsorption capacity of the PBAT-Z-6M membrane as a function of time is shown in Figure 5. The adsorption capacity of both contaminants increases relatively fast in the first ~120 min. Then, it started to slow down until it reached an equilibrium value at ~1440 min of exposure. Pseudo-first-order (PFO) and pseudo-second-order (PSO) equations allow explaining the adsorption kinetics of several systems [28]. PFO model assumes that the adsorption rate is proportional to the difference between the adsorbed concentration and the number of available sites, which is an accurate approximation of a high starting solute concentration. On the other hand, PSO model considers second-order proportionality assuming that the rate-limiting step involves chemisorption. The integral form of PFO and PSO model can be expressed according to Equations (3) and (4), respectively:(4)$$q(t)=q_{e}[1-exp(-k_{1}\;t)]$$
(5)$$q(t)=\frac{q_{e}^{2}\;k_{2}\;t}{1+q_{e}\;k_{2}\;t}$$
where *t* is the contact time (min), *q_t_* and *q_e_* are the adsorption capacity at a time *t* and at equilibrium (mg/g), *k*_1_ (min^−1^) is the pseudo-first-order rate constant, and *k*_2_ (g/mg min) is the pseudo-second-order rate constant.

Experimental data of adsorption capacity as a function of time were fitted according to the PFO and PSO equations. The best curve was also plotted in Figure 5, and the resulting parameters are shown in Table 3. As shown in Figure 5, the PSO equation represents the adsorption kinetic of both contaminants more accurately, which is also reflected in the higher coefficient of determination, R^2^. According to the model, this result indicates that chemisorption is the primary mechanism involved in the adsorption of pollutants. Adsorption of MB and TC by zeolite-based adsorbent is consistent with PSO kinetics [60,62]. Additionally, it can be observed that the equilibrium adsorption capacity of the membrane against TC was approximately 6-fold higher than that of MB, which reflects a higher removal efficiency of TC, in agreement with the discussion about adsorption isotherms.

#### 2.3.4. Effect of pH and Reusing

The effect of the pH on the adsorption capacity of the membrane is presented in Figure 6. The higher adsorption capacity was found at neutral pH values (5–8) for TC, showing a pH-dependent removal process. This behavior is due to the TC speciation in an aqueous solution. For a pH < 3.30, TC predominates in its protonated form (TCH_3_^+^), while the zwitterion TCH_2_^+ −^ is the predominant species from pH 3.30 to 7.70. In an alkaline medium, the main species are TCH^-^ for a pH higher than 7.70 and TC^2−^ for a pH > 10 [59,63]. The zwitterion is the only sorptive species in the pH range of 5.50–7.50. It is preferably adsorbed onto the zeolite surface through cation exchange between TCH_2_^+ −^ species and positive sites in the zeolite framework (i.e., Na^+^) [58,63]. In addition, hydrogen bonding between ionized TC molecules and surface silanol groups has been reported as a possible adsorption route, as well as the interaction of Lewis acid sites of zeolites with the adsorbate [64]. The high removal efficiency of TC in weak acidic or slightly basic environments has also been reported by thermally treated clinoptilolite [60] and by iron dope zeolite [59,63], showing that the adsorption of zwitterionic TC is favored over other adsorbate species. At a pH above 8, adsorption capacity decreased from electrostatic repulsion between anionic TC species and Si-O^−^ surface groups formed through deprotonation of clinoptilolite (reaction 2).

On the other hand, the adsorption capacity for MB increases with an increase in pH, showing better performance at a basic pH (7–10). MB presents a pK_a_ of 3.80 [9], indicating that for a pH > 4, MB exists as a molecular cation in water. The enhancement in the MB removal capacity as the pH raised from 3.50 to 10.0 is related to the electrostatic interaction between MB cation and surface silanol groups onto the clinoptilolite zeolite. At acidic pH, the adsorption mechanism is dominated by repulsive interaction between the protonated groups (Si-OH_2_^+^) and cationic MB. As the pH is further raised, Si-O^−^ groups are produced in the zeolite surface due to a deprotonation reaction. Thus, attractive electrostatic interaction will favor the adsorption of MB cation. Additionally, MB adsorption can be performed by cationic exchange and coordination with the oxygen donor atoms of the zeolite surface [48,65]. This mechanism may explain the low removal capacity under acidic conditions, where repulsive interactions predominate, limiting MB adsorption.

The efficiency of contaminants removal after five cycles is shown in Figure 7. The efficiency of MB removal was reduced to ~75% and ~50% in the second and third cycles, respectively. In contrast, the efficiency of TC removal was reduced to ~50% and ~10% in the same number of cycles. The removal of both contaminants after 4 cycles was reduced to less than 10%. These results show that the PBAT-Z membrane partially preserves its adsorption capacity after reuse. Notably, the efficiency of MB removal was maintained at higher values than TC removal. The regeneration depends on the treatment applied to the adsorbent after the remediation process. In our case, the regeneration of the PBAT-Z-6M adsorbent was carried out in deionized water. It has been reported that the efficiency of powdered zeolites to remove TC reaches ~40% on the second cycle after an alkaline treatment with 0.01 M NaOH [66]. Compared with the present investigation’s results, ~13% higher efficiency was achieved after the second reuse cycle. Higher efficiencies in adsorbent regeneration can be obtained by using severe treatments. For the removal of ciprofloxacin by clinoptilolite zeolite, it has been shown that the adsorption capacity of zeolite recovered up to 70% after treatment with surface dielectric barrier discharge plasma [22]. Other treatments for the regeneration of zeolite include high-temperature calcination and Fenton oxidation [65]. Nevertheless, the adsorption capacity for MB was reduced 40% compared to the fresh adsorbent.

## 3. Materials and Methods

### 3.1. Materials

PBAT (Ecoworld^®^-JinHui ZhaoLong High Technology Co., Shanxi Province, China) and clinoptilolite zeolite (ZEOCOL S.A.S, Quindío, Colombia) were used to prepare the membranes. Before experiments, clinoptilolite was milled and sieved through a #300 mesh to obtain a particle size ≤ 44 µm. Hydrochloric acid (36.5% wt.), chloroform, rhodamine B, and methylene blue were supplied by Cicarelli (Santa Fe, Argentina). N,N-dimethylformamide (DMF) was purchased from Merck (Darmstadt, Germany). Tetracycline hydrochloride was supplied from Saporiti (Buenos Aires, Argentina). All these chemicals have a purity of 98% or higher. Deionized water was used to prepare reagents.

### 3.2. Fabrication of Membranes

A PBAT solution for electrospinning (12.6% *w*/*w*) was prepared by dissolving the polymer in a chloroform/DMF solution (80:20 *v*/*v*) using a magnetic stirrer for 2 h, based on the reported methodology [67]. Clinoptilolite zeolite (1% *w*/*w* PBAT) was dispersed in 2 g of DMF using a vortex and incorporated into the polymer solution under vigorous stirring for 1 h. The obtained solution was electrospun using the homemade apparatus described previously [68]. Briefly, the process consisted in placing the solution in a plastic syringe and expelling it by six needles (inner diameter = 0.8 mm) at a controlled feed rate of 5.5 mL/h. A cylindrical collector (diameter = 6 cm) was located at a distance of 11.5 cm, and a voltage of 30 kV was applied between the needles and the collector.

To modify and tune the materials’ water wettability, the PBAT-Z electrospun membranes were immersed in HCl aqueous solutions of different concentrations (1 M, 3 M, and 6 M) for 24 h under agitation, according to the following procedure [41]. As a control, zeolites and a PBAT membrane (without zeolites) were acid-treated with HCl 6 M. Samples were washed three times with distilled water to remove the chloride ions from the HCl, and then dried at 60 °C overnight [52]. The terminology of acid-treated samples includes the molarity of the solution used (for example, PBAT-Z-3M refers to the PBAT-Z membrane after being immersed for 24 h in HCl 3 M).

### 3.3. Characterizations

#### 3.3.1. Microscopy

The morphology and elemental composition of electrospun membranes were analyzed using a scanning electron microscope (SEM) with a field emission gun (ZeissDSM982 Gemini, Oberkochen, Germany) operated at 30 kV and equipped with an energy dispersive spectrometer (EDS) (Oxford Analysis). Samples were sputtered with a thin platinum layer before observation and placed on an aluminum holder using carbon tape for analysis.

Fluorescence microscopy was used to determine the dispersion of zeolite in PBAT membranes. The electrospun membrane was immersed in an aqueous rhodamine B solution (2 mg/L, pH = 6) for 24 h, following the experimental procedure described in reference [69]. The ratio between zeolite and rhodamine B solution was 90 mg/L. The sample mat was recovered from the solution, dried at 60 °C for 12 h, and analyzed using a multiphoton confocal microscope model LSM 980 equipped with a 543 nm laser excitation source (Carl Zeiss, Oberkochen, Germany).

#### 3.3.2. Infrared Spectroscopy

Infrared spectra were obtained using a Fourier Transform Infrared Spectrometer (Jasco FT-IR 4100, Tokio, Japan) equipped with an attenuated total reflectance module (ZnSe crystal). The spectra were recorded in a range from 4000 to 600 cm^−1^, at a resolution of 4 cm^−1^, averaging 24 scans per sample. The spectra were normalized to the band at 1710 cm^−1^, which is expected not to change after acid treatment.

#### 3.3.3. Contact Angle

The contact angle was measured using an Optical Tensiometer (OneAttension theta—Biolin Scientific, Gothenburg, Sweden). Drops of distilled water (10 µL) were placed on different zones of the membrane’s surface, and at least 10 determinations were performed to calculate the average value.

#### 3.3.4. Adsorption Experiments

Preliminary batch experiments were performed to evaluate the performance of PBAT membranes (PBAT and PBAT-Z) and power zeolite before and after acid treatment. Adsorption experiments were carried out by mixing 45 mg of the adsorbent and 5 mL aliquots of TC or MB aqueous solutions at concentration of 25 mg/L and 2 mg/L, respectively, using 25 mL Erlenmeyer flasks. The flask was stirred in an orbital shaker at 120 rpm for 24 h. The pH of solutions was fixed at 6.30. The residual TC and MB concentrations were determined using an UV–Vis spectrometer (Shimadzu UV-1800, Kyoto, Japan) at a wavelength of 357 nm and 664 nm, respectively. Calibration curves were done using aqueous solutions in the range of 0.5 to 200 mg/L for TC and 0.25 to 8 mg/L for MB. The glassware used through all experiments was covered with aluminum foil to prevent the photolysis of TC. All experiments were performed in triplicate and at room temperature (25 ± 0.1 °C). Calibration curves for UV–Vis spectrometer measurements were done by preparing TC and MC stock solutions. TC stock solution (1000 mg/L) was prepared daily by dissolving 100 mg of TC in deionized water to a final volume of 100 mL. A methylene blue (MB) stock solution (1000 mg/L) was prepared by dissolving 250 mg of MB in deionized water up to a final volume of 250 mL. The calibration curves and solutions used in the adsorption experiments were prepared by diluting the stock solution of each pollutant considered (MB or TC) to obtain the different concentrations analyzed in this work.

According to the preliminary results obtained at 24 h (described above), the highest adsorption for both contaminants was obtained for the mats treated with HCl 6 M. Therefore, further adsorption isotherm, kinetics, and reuse experiments were performed with the PBAT-Z-6M mat.

In the adsorption experiments, 5 mL aliquot of adsorbate solution at different initial concentrations were mixed with 45 mg of PBAT-Z-6M membrane and shaken for 24 h to ensure it reaches equilibrium state. The concentrations were varied in the 1–200 mg/L range for TC and in the 1–8 mg/L range for MB.

In the kinetic studies, 45 mg of PBAT-Z-6M membrane and 5 mL of TC (12.5 mg/L) or MB (1 mg/L) were added to separate 25 mL Erlenmeyer flasks and shaken at 120 rpm for a time interval between 15 min and 24 h. Then, the residual concentration in each solution after a given time was measured.

The adsorption capacity *q* (mg/g) after a time *t* (*q_t_*) was calculated by Equation (6):
(6)$$q_{t}=\frac{C_{o}-C_{t}}{m}V$$
where *C_o_* is the initial concentration (mg/L), *C_t_* is the concentration (mg/L) at time *t* (min), *m* (g) is the mass of natural zeolite in the membrane, and *V* (L) is the volume of solution.

The effect of pH on the adsorption was studied in the range of 3 to 10. The pH was adjusted by adding 0.1 M HCl or 0.1 M NaOH to the adsorbate solutions.

The reuse of the membrane was evaluated by measuring the adsorption capacity in several cycles with a fixed initial concentration of the contaminant (TC: 50 mg/L; MB: 2 mg/L) and using 45 mg of adsorbent per 5 mL of contaminant solution. Each cycle consists of the adsorption of the contaminant (24 h under stirring) and washing of the membrane (24 h in distilled water under agitation). The efficiency of each cycle is then obtained using Equation (2):(7)$$Efficiency(\%)=100\;\frac{C_{n}}{C_{1}},$$
where *C_n_* (mg/L) is the final concentration after *n* adsorption cycles, and *C*_1_ (mg/L) is the final concentration after the first adsorption cycle.

## 4. Conclusions

A PBAT mat with a natural clinoptilolite zeolite immobilized between its nanofibers showed excellent removal capacity for tetracycline (~800 mg/g) and methylene blue (~100 mg/g) from water, surpassing the performance of other zeolite-based adsorbents reported in the literature.

The high adsorption capacity achieved resulted from two strategies followed in the production of the membrane—the electrospinning of a zeolite dispersion in a polymeric solution with the chemical affinity between them, and the transformation of the electrospun nanofiber surface from hydrophobic to hydrophilic by applying an acid treatment. The tuned material presented enhanced the interaction with water and, therefore, the possibility of adsorption of contaminants.

The adsorption mechanism of the system adsorbent-adsorbate comprises cation exchange and coordination reactions compatible with pseudo-second-order kinetics. The developed nanocomposite has several comparative advantages over other adsorbents. Thanks to the adsorption mechanism and the charge taken by MB and TC, its adsorptions are very high at pH 7. Furthermore, since the adsorbent is included in an extensive, self-supporting material, it can be removed from the water without generating additional contamination, remove the contaminant and reuse the adsorbent. Finally, although the research proposed here does not focus on this application, the entangled nanofiber structure can be used as a filtration membrane. Overall, it contributes to developing advanced electrospun nanofibers for the remediation of emergent contaminants.

## Figures and Tables

**Figure 1 molecules-28-00081-f001:**
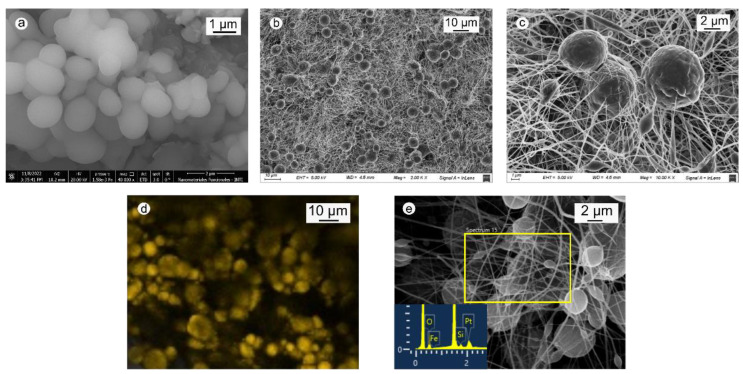
(**a**) SEM images of zeolites; (**b**) PBAT-Z membrane at a magnification 2 kX; (**c**) same in (**b**) at a magnification 10 kX; (**e**) EDS analysis of materials; (**d**) fluorescence microscopy images of PBAT-Z membranes dyed with rhodamine B; and I EDS spectrum of PBAT-Z-6M membranes.

**Figure 2 molecules-28-00081-f002:**
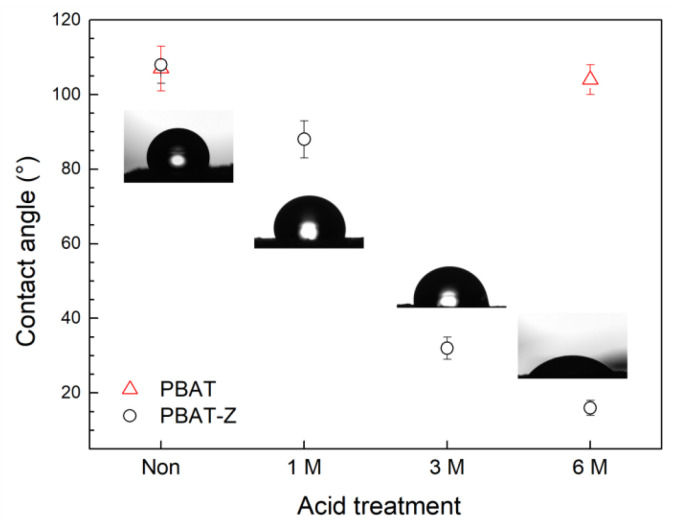
Contact angle of the membranes treated with different acid concentrations, and representative images of drops on PBAT-Z and PBAT-Z-6M membranes.

**Figure 3 molecules-28-00081-f003:**
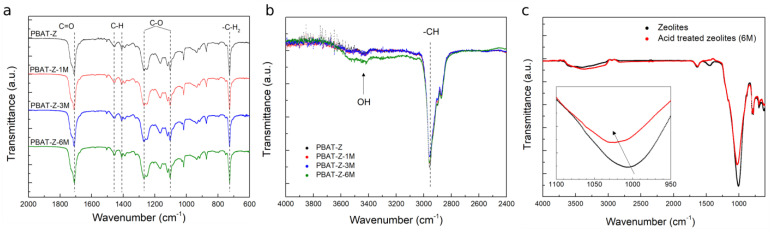
FTIR spectra of PBAT-Z membranes treated with different concentration of HCl in the range 2000–600 cm^−1^ (**a**) and 4000–2400 cm^−1^ (**b**). (**c**) FTIR spectra corresponding to zeolites before/after the acid treatment at higher HCl concentration.

**Figure 4 molecules-28-00081-f004:**
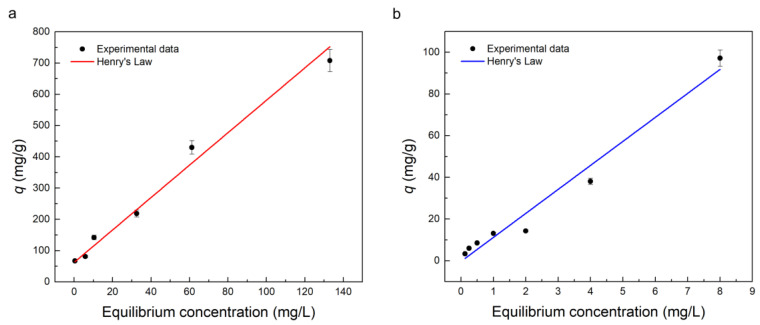
Adsorption isotherm of PBAT-Z-6M membrane against TC (**a**) and MB (**b**). The filled line shows the fitting using Henry’s model. The adsorbent dose was 90 mg/L for both contaminants.

**Figure 5 molecules-28-00081-f005:**
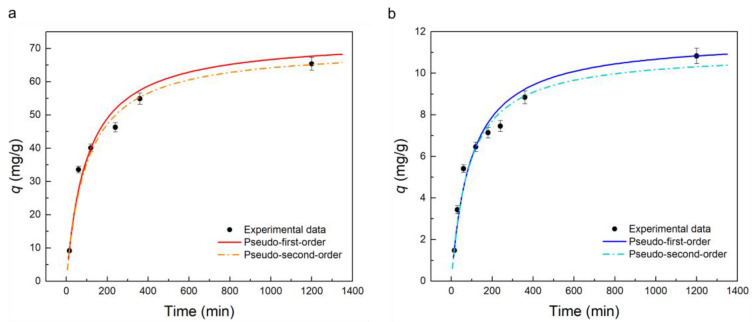
Adsorption kinetics of PBAT-Z-6M membrane against TC (**a**) and MB (**b**), and non-linear fitting using pseudo-first-order and pseudo-second-order models. The adsorbent dose was 90 mg/L for both contaminants.

**Figure 6 molecules-28-00081-f006:**
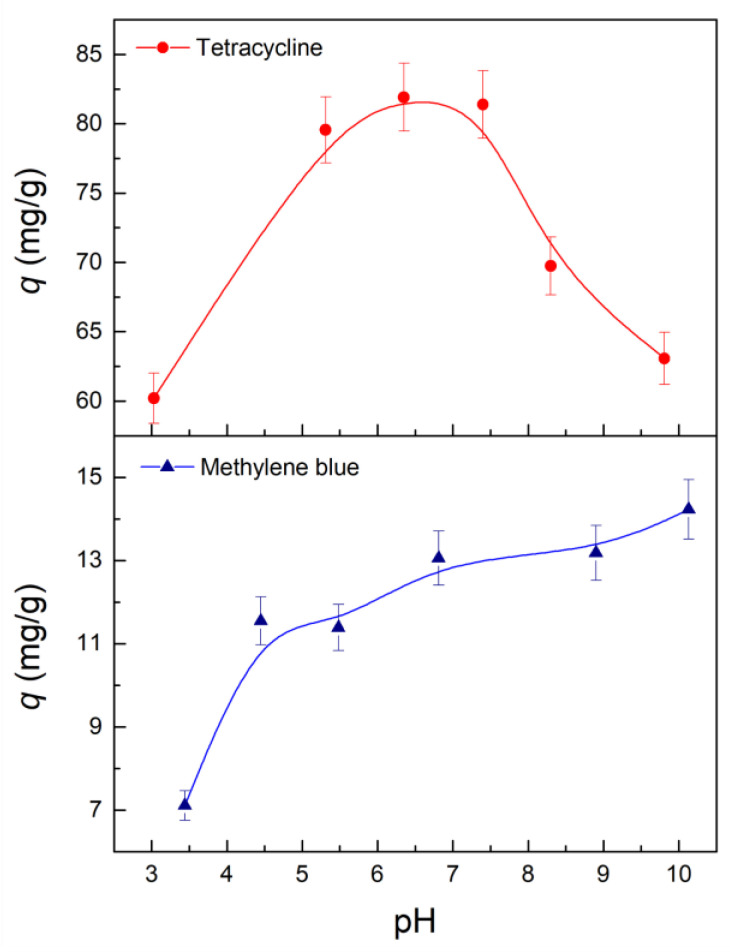
Effect of pH on the adsorption capacity of PBAT-Z-6M membrane against TC and MB. Lines are only visual guides.

**Figure 7 molecules-28-00081-f007:**
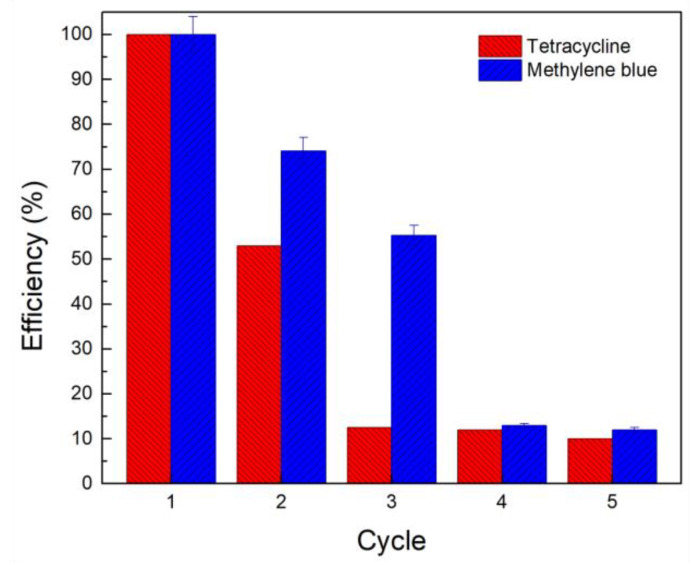
Adsorption efficiency of the PBAT-Z-6M membrane against tetracycline and methylene blue as a function of reuse cycles.

**Table 1 molecules-28-00081-t001:** Adsorption capacities for PBAT and PBAT-Z membranes compared with that of powdered zeolite against TC and MB.

Material	*q* TC (mg/g) ^1^	*q* MB (mg/g) ^2^
PBAT	3.1 ± 0.5	0.11 ± 0.3
PBAT-6M	3.5 ± 0.7	0.13 ± 0.3
PBAT-Z	37.9 ± 2.7	8.2 ± 0.5
PBAT-Z-6M	142.4 ± 7.8	12.2 ± 0.7
Powdered zeolite	27.3 ± 3.8	3.3 ± 0.4
Powdered zeolite-6M	50.7 ± 3.0	4.6 ± 0.2

^1^ *C_o_* = 25 mg/L at pH 6.30. ^2^ *C_o_* = 2 mg/L at pH 6.30.

**Table 2 molecules-28-00081-t002:** Fitting parameters of the adsorption isotherms using Henry’s model.

Contaminant	Henry’s Constant (L/g)	R^2^
TC	5.18 ± 0.51	0.962
MB	11.51 ± 0.87	0.972

**Table 3 molecules-28-00081-t003:** Adjustment parameters obtained for the adsorption kinetic curves using pseudo-first-order and pseudo-second-order models. Both non-linear fittings were performed considering the experimental data between 0 and 1200 min.

Adsorbate	Pseudo First Order			Pseudo Second Order		
	*k_1_* (min^−1^)/(10^−3^)	*q_e_* (mg/g)	R^2^	*k_2_* (g/mg min)/(10^−5^)	*q_e_* (mg/g)	R^2^
TC	12.4 ± 2.0	55.3 ± 4.4	0.911	14.4 ± 1.3	70.5 ± 2.1	0.998
MB	13.1 ± 2.0	8.68 ± 0.64	0.887	104 ± 11	11.06 ± 0.77	0.994

## Data Availability

Not applicable.

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
