# Peer review of "Adsorption of Methylene Blue and Tetracycline by Zeolites Immobilized on a PBAT Electrospun Membrane"

_molecules, 2022, doi:10.3390/molecules28010081_

Round 1

Reviewer 1 Report

The manuscript by Goyanes et al. reports the adsorption of Methylene Blue and Tetracycline by Zeolites Immobilized on a PBAT Electrospun Membrane.

Although the work is interesting, some aspects should still be considered to improve the quality of this work.

1. The paper should be written in the third person.

2. In some parts English must be improved.

3. The style of writing the titles and subtitles must be unified. The same applies to the numbers of equations.

3. The marks, especially the q must be in italic.

4. Page 5, line 183, C-H2 2 must be in subscript.

5. Page 12, line 433, 60°C.

6. Page 13, line 470, equation 5 not equation 1.

7. Page 13, line 478, equation 6 not equation 2.

8. The references should be checked again and written according to the Instructions for the Authors.

9. Why did the Authors selected the Henry`s adsorption model? This model is usually applied for the adsorption of gases. Why didn’t they use the Langmuir or Freundlich isotherm models? The Authors didn’t present the Henry`s adsorption model nowhere in the manuscript (teorethical background with the formula). Please do so.

10. Please improve the 3. Materials and Methods/ 3.3.4 Adsorption Experiments section with more details. The Authors didn’t explain at which concentrations they carried out the kinetic experiments.

Overall, it is an interesting study and should be considered for publication once the issues have been resolved. I recommend major revision of the manuscript before publication.

Author Response

The manuscript by Goyanes et al. reports the adsorption of Methylene Blue and Tetracycline by Zeolites Immobilized on a PBAT Electrospun Membrane.

Although the work is interesting, some aspects should still be considered to improve the quality of this work. 

We thank the reviewer for thinking our manuscript is interesting.

Point 1. The paper should be written in the third person.

Response 1: Thank you for your suggestion.  We have written the paper in the third person.

Point 2. In some parts English must be improved.

Response 2: The English of the manuscript has been improved, correcting various errors and inconsistencies.

Point 3. The style of writing the titles and subtitles must be unified. The same applies to the numbers of equations.

Response 3: As suggested by the reviewer, the writing style of titles, subtitles and equations was unified based on the format of the journal.

Point 4. The marks, especially the q must be in italic.

Response 4: Thank you for pointing this out; the changes have been made.

Point 5. Page 5, line 183, C-H2 2 must be in subscript.

Response 5: We appreciate the corrections of the reviewer. C-H2 has been changed to C-H2.

Point 6. Page 12, line 433, 60°C.

Response 6: We appreciate the correction, 60ºC was changed to 60 °C

Point 7. Page 13, line 470, equation 5 not equation 1.

Response 7: This error has been corrected.

Point 8. Page 13, line 478, equation 6 not equation 2.

Response 8: Thanks for your comment. This error has been corrected.

Point 9. The references should be checked again and written according to the Instructions for the Authors.

Response 9: As suggested by the reviewer, the references have been checked and were written according to journal format.

Point 10. Why did the Authors selected the Henry`s adsorption model? This model is usually applied for the adsorption of gases. Why didn’t they use the Langmuir or Freundlich isotherm models? The Authors didn’t present the Henry`s adsorption model nowhere in the manuscript (teorethical background with the formula). Please do so.

Response 10: We appreciate the comments of the reviewer. The Henry model, as well as most of the models used for adsorption, has been primarily developed for the adsorption of gases. As it is very well-known, applying these models to liquid phases requires converting the partial pressures to concentrations. In addition, the Henry model was already used in the literature to describe liquid-phase adsorption [see, for instance,1-4].

Henry´s model is the most straightforward adsorption model in which the amount of adsorbate is proportional to the concentration. This model describes the adsorption in the linear range, usually observed at low concentrations. In the studied range, our results revealed an isotherm with a clearly linear behavior; therefore, Henry´s model applies to our case but not Langmuir or Freundlich models.

Nevertheless, Langmuir and Freundlich´s fits were performed just for reviewer purposes, and their results are presented below. As seen in the table, the parameters of the fits and the statistics have a significant error mainly because we are analyzing the low-concentration adsorption range of the system.

Tetracycline

Methylene blue

Henry’s

Freundlich model

Langmuir model

Henry’s constant (L/g)

R2

KF (mg/g)

N

R2

???

KL (L/g)

qm (mg/g)

R2

???

TC

5.18±0.51

0.96

28±11

1.54±0.22

0.91

13.9

8.1±6.5

1300±700

0.86

28.1

MB

11.51±0.87

0.97

10.6±2.5

0.99±0.14

0.89

2.03

Do not converge

To clarify the reasons for not including these adjustments in the paper, the following sentences have been added to the manuscript, in lines 279-293:

“The adsorption isotherms were obtained by representing the adsorption capacity (q) against equilibrium concentration at 24 h. Figure 4 displays the adsorption isotherms of the PBAT-Z-6M membrane for TC (a) and MB (b), showing a linear behaviour for both contaminants in the studied range. Henry's model is the most straightforward adsorption model in which the amount of adsorbate is proportional to the concentration, according to equation 3:

qe=KHCe

(3)

where qe (mg/g) is the amount of adsorption at equilibrium, KH (L/g) is Henry's constant, and Ce (mg/L) is the equilibrium concentration of adsorbate in solution. This model describes the adsorption usually at low concentrations, as those analysed in this work. For the same reason, Langmuir's or Freundlich's models do not apply to the present case. In addition, Henry's model has already been used in literature to describe liquid-phase adsorption. For example, the adsorption of heavy metals [1,2], antibiotics [3] and dyes [4] from water by the solid adsorbent.”

References

[1] Frantz, T.S.; Silveira, N.; Quadro, M.S.; Andreazza, R.; Barcelos, A.A.; Cadaval, T.R.S.; Pinto, L.A.A. Cu(II) Adsorption from Copper Mine Water by Chitosan Films and the Matrix Effects. Environ. Sci. Pollut. Res. 2017, 24, 5908–5917, doi:10.1007/S11356-016-8344-Z/FIGURES/7.

[2] Chen, X.; Hossain, M.F.; Duan, C.; Lu, J.; Tsang, Y.F.; Islam, M.S.; Zhou, Y. Isotherm Models for Adsorption of Heavy Metals from Water - A Review. Chemosphere 2022, 307, 135545, doi:10.1016/J.CHEMOSPHERE.2022.135545.

[3] Li, R.; Zhang, Y.; Chu, W.; Chen, Z.; Wang, J. Adsorptive Removal of Antibiotics from Water Using Peanut Shells from Agricultural Waste. RSC Adv. 2018, 8, 13546–13555, doi:10.1039/C7RA11796E.

[4] You, H.; Huang, B.; Cao, C.; Liu, X.; Sun, X.; Xiao, L.; Qiu, J.; Luo, Y.; Qian, Q.; Chen, Q. Adsorption–Desorption Behavior of Methylene Blue onto Aged Polyethylene Microplastics in Aqueous Environments. Mar. Pollut. Bull. 2021, 167, 112287, doi:10.1016/J.MARPOLBUL.2021.112287.

Point 11. Please improve the 3. Materials and Methods/ 3.3.4 Adsorption Experiments section with more details. The Authors didn’t explain at which concentrations they carried out the kinetic experiments.

Response 11: We appreciate the reviewer's comments. We agree that the Material and Methods section was not described sufficiently to allow others to replicate it. The entire Material and Methods section was improved by adding experimental details and references used in the adsorbents synthesis to further clarify this issue.

In the section “3.3.4 Adsorption experiment”, we have included the concentration at which kinetic experiments were performed (see lines 503-506 of the revised version of manuscript):

“In the kinetic studies, 45 mg of PBAT-Z-6M membrane and 5 mL of TC (12.5 mg/L) or MB (1 mg/L) were added to separate 25 mL Erlenmeyer flasks and shaken at 120 rpm for a time interval between 15 min and 24 h. Then, the residual concentration in each solution after a given time was measure.”.

Overall, it is an interesting study and should be considered for publication once the issues have been resolved. I recommend major revision of the manuscript before publication.

Reviewer 2 Report

Dear Authors

Many thanks for your valuable research.

1.Please moved materials and methods section to the section 2.

2. What is novelty of this study?

3. Please write the keywords in alphabetical order.

4. In the introduction, similar adsorbents that have been synthesized should be mentioned and their disadvantages compared to the adsorbents of this study should be mentioned.

5. Please cite to synthesis method references of adsorbents.

6. Please use the following articles in your manuscript:

1.      Hassan Hashemi, Shima Bahrami, Zahra Emadi, Hamideh Shariatipor, Majid Nozari. Optimization of ammonium adsorption from landfill leachate using
montmorillonite/hematite nanocomposite: response surface method
based on central composite design. Desalination and Water Treatment 232 (2021) 39–54

2.      Vahideh Parvaresh, Hassan Hashemi, Abbas Khodabakhshi, Morteza Sedehi. Removal of dye from synthetic textile wastewater using agricultural wastes and determination of adsorption isotherm. Desalination and Water Treatment. 111 (2018) 345–350.

Best regards,

Author Response

Dear Authors

Many thanks for your valuable research.

We thank the reviewer for thinking our manuscript is interesting.

Point 1. Please moved materials and methods section to the section 2.

Response 1: We appreciate the reviewer's suggestion. The Material and Method section was placed in section 3 of the manuscript following the journal’s template and instruction for authors.

Point 2. What is novelty of this study?

Response 2: We thank the reviewer for deep reading our manuscript and providing an honest opinion. Reading the reviewer´s opinion, it is clear that the purpose and significance were not clearly explained in the first version of our manuscript. To highlight the novelty of the work, we have added the following sentence in the introduction (see lines 132-136):

“Even though zeolites are often used in combination with other polymers [42,43], there are no precedents in which they are combined with PBAT nanofibers as a strategy to ensure optimal zeolite dispersion for use in removing water contaminants. Also, acid treatments to improve the interaction with water of PBAT-zeolites nanofibers have not been reported either.”

Point 3. Please write the keywords in alphabetical order.

Response 3: As suggested by the reviewer, the keywords were written in alphabetical order in the revised version of the manuscript.

Point 4. In the introduction, similar adsorbents that have been synthesized should be mentioned and their disadvantages compared to the adsorbents of this study should be mentioned.

Response 4: There are no reports in the literature of PBAT electrospun filled with zeolites for its application for water treatment. Therefore, our work represents the first study of this type. On the other hand, there are a few previous reports specifically for PBAT-zeolite composites but focused on other applications.

This point was rewritten and highlighted in the introduction section (see lines 132-136):

“Even though zeolites are often used in combination with other polymers [42,43], there are no precedents in which they are combined with PBAT nanofibers as a strategy to ensure optimal zeolite dispersion for use in removing water contaminants. Also, acid treatments to improve the interaction with water of PBAT-zeolites nanofibers have not been reported either.”

And in the new version of the manuscript, we compared our results and advantages in the discussion section, in lines 309-311:

For example, it was reported an adsorption capacity of ~255 mg/g for zeolitic imidazolate framework nanoparticles/polyacrylonitrile nanofiber membrane (ZIF/PAN) [42]…”

Point 5. Please cite to synthesis method references of adsorbents.

Response 5: Thank you for pointing out this issue. In the Material and method section, we have included references to the synthesis methods and experimental procedure followed throughout this work. In addition, the Material and Methods section was rewritten to include the details of synthesis procedures.

Point 6. Please use the following articles in your manuscript:

  1. Hassan Hashemi, Shima Bahrami, Zahra Emadi, Hamideh Shariatipor, Majid Nozari. Optimization of ammonium adsorption from landfill leachate using
    montmorillonite/hematite nanocomposite: response surface method
    based on central composite design. Desalination and Water Treatment 232 (2021) 39–54
  2. Vahideh Parvaresh, Hassan Hashemi, Abbas Khodabakhshi, Morteza Sedehi. Removal of dye from synthetic textile wastewater using agricultural wastes and determination of adsorption isotherm. Desalination and Water Treatment. 111 (2018) 345–350. 

Response 6: Those articles are very interesting with high scientific value. We have included, in the introduction section, the removal of ammonium ions by montmorillonite/hematite nanocomposite, and the adsorption of dyes by biomass-derived adsorbents (see lines 91-93):

“Recently, there has been increasing interest in removing emerging pollutants from water using naturally occurring adsorbents such as clay minerals [18,19], biomass [20,21] and zeolites [22,23].”

Reviewer 3 Report

In this study, the authors we prepared a polybutylene adipate terephthalate (PBAT) nanofibrous membrane incorporating clinoptilolite zeolite and tested it in the adsorption of methylene blue (MB) and tetracycline (TC) in water. They reported that the acid treatment to the adsorbent considerably increased its adsorption capacity. This is a well-written work in a clear and easy to follow way that deserves to be published after the following minor corrections:

1. Modify the last sentence of the Abstract to be more accurate as cyclic experiments showed a significant decrease in adsorption capacity after each cycle.

2. Eliminate first-person language throughout the text.

3. Provide definition for all quantities presented in equations along with their units.

4. Paragraph 3.3.4: Provide the volume of solutions tested.

5. It is mentioned that adsorption isotherms were obtained by varying the initial concentration of TC between 1-200 mg/L and that the maximum adsorption capacity achieved was around 800 mg TC/g zeolite, but in Fig. 4a the concentration range shown is 1-140 mg/L and the maximum adsorption capacity around 800 mg TC/g.

6. Line 470: Correct “equation 1” to “equation 5”.

7. 10% of references at least are self-citations.

Author Response

In this study, the authors we prepared a polybutylene adipate terephthalate (PBAT) nanofibrous membrane incorporating clinoptilolite zeolite and tested it in the adsorption of methylene blue (MB) and tetracycline (TC) in water. They reported that the acid treatment to the adsorbent considerably increased its adsorption capacity. This is a well-written work in a clear and easy to follow way that deserves to be published after the following minor corrections:

We thank the opinion of the reviewer about our work.

Point 1. Modify the last sentence of the Abstract to be more accurate as cyclic experiments showed a significant decrease in adsorption capacity after each cycle.

Response 1: The last sentence in the abstract has been modified as suggested by the reviewer, in lines 37-38:

“The material can be reused after removal without generating additional contamination, although losing some effectivity.

Point 2. Eliminate first-person language throughout the text.

Response 2: Thank you for your suggestion, we have written the paper in the third person.

Point 3. Provide definition for all quantities presented in equations along with their units.

Response 3: As the reviewer suggested, the quantities presented in the equations and their units were clearly defined throughout the text.  

Point 4. Paragraph 3.3.4: Provide the volume of solutions tested.

Response 4: The volume of the solution used in the adsorption and kinetic experiments was included in the revised version of the manuscript, in lines 486-488:

“Adsorption experiments were carried out by mixing 45 mg of the adsorbent and 5 mL aliquots of TC or MB aqueous solutions at concentration of 25 mg/L and 2 mg/L, respectively, using 25 mL Erlenmeyer flasks.”

Point 5. It is mentioned that adsorption isotherms were obtained by varying the initial concentration of TC between 1-200 mg/L and that the maximum adsorption capacity achieved was around 800 mg TC/g zeolite, but in Fig. 4a the concentration range shown is 1-140 mg/L and the maximum adsorption capacity around 800 mg TC/g.

Response 5: Thank you for pointing out this issue. We agree this point was not clear enough in the first version of our manuscript. Adsorption isotherms were carried out by varying the initial concentration from 1 to 200 mg/L. Still, in Figure 4, adsorption capacity (q) is plotted against equilibrium concentrations, i.e., the residual concentration found in solution after 24 h of adsorption by PBAT-Z-6M. As the reviewer knows, this is the usual procedure for representing the performance of an adsorbent. In addition, q is the ratio of the tetracycline mass adsorbed per mass of adsorbent. Hence, given the small concentration used in the experiments, it is expected to have a high adsorption capacity. 

The following sentence was added to the manuscript to clarify this issue further (See lines 279 to 280):

“The adsorption isotherms were obtained by representing the adsorption capacity (q) against equilibrium concentration at 24 h.”

Point 6. Line 470: Correct “equation 1” to “equation 5”.

Response 6: The equation´s numbers have been correctly numbered in the new version of the manuscript.

Point 7. 10% of references at least are self-citations.

All the references are appropriate and necessary to properly describe our work. In any case, in the new version of our manuscript, we have included further references from other authors as suggested by other reviewers.

Round 2

Reviewer 1 Report

I am very grateful to the Authors for taking my comments into consideration.

Please correct the following:

1. Page 3, line 104, C should not be capital.

2. Page 3, line 114, remove the extra dot.

3. Page 3, line 148, remove the dot after 2.1

Remove the dots at the end of the subtitles

4. Page 7, line 287, only q should be italic. The mark in subscript as well as the units should not be italic so please correct it through the manuscript, and the figures as well.

5. Page 7, line 292, to should be written instead of yo.

6. Page 9, line 348, put the dot at the end of the sentence.

7. Page 14, line 485, powder zeolite?

8. Page 14, line 509, was measured.

9. Page 14, line 512, comma before Ct.

10. Please explain in the 3. Materials and Methods/ 3.3.4 Adsorption Experiments section how did you prepare the solution of TC for determination on the UV-Vis spectrometer.

Overall, it is an interesting study and should be considered for publication once the issues have been resolved. I recommend minor revision of the manuscript before publication.

Author Response

Reviewer 1

I am very grateful to the Authors for taking my comments into consideration. Please correct the following:

  1. Page 3, line 104, C should not be capital.

This has been modified.

  1. Page 3, line 114, remove the extra dot.

The extra dot was removed in the new version of the manuscript.

  1. Page 3, line 148, remove the dot after 2.1 and remove the dots at the end of the subtitles

As suggested by the reviewer, all the dots have been removed.

  1. Page 7, line 287, only q should be italic. The mark in subscript as well as the units should not be italic so please correct it through the manuscript, and the figures as well.

All “q” letters are now in italics, and the units are in regular letters.

  1. Page 7, line 292, to should be written instead of yo.

This mistake has been corrected.

  1. Page 9, line 348, put the dot at the end of the sentence.

We have added the dot at the end of this line.

  1. Page 14, line 485, powder zeolite?

The reviewer is correct. Instead of “powder zeolite”, we have written “powdered zeolite” in the new version of the manuscript.

  1. Page 14, line 509, was measured.

We have changed “measure” by “measured”.

  1. Page 14, line 512, comma before Ct.

The comma has been added.

  1. Please explain in the 3. Materials and Methods/ 3.3.4 Adsorption Experiments section how did you prepare the solution of TC for determination on the UV-Vis spectrometer.

This information has been added in the new version of the manuscript.

Calibration curves for UV-VIs spectrometer measurements were done by preparing TC and MC stock solutions. TC stock solution (1000 mg/L) was prepared daily by dissolving 100 mg of TC in deionized water to a final volume of 100 mL. A methylene blue (MB) stock solution (1000 mg/L) was prepared by dissolving 250 mg of MB in deionized water up to a final volume of 250 mL. The calibration curves and solutions used in the adsorption experiments were prepared by diluting the stock solution of each pollutant considered (MB or TC) to obtain the different concentrations analyzed in this work.

Overall, it is an interesting study and should be considered for publication once the issues have been resolved. I recommend minor revision of the manuscript before publication.

We thank the opinion of the reviewer about our work.

Reviewer 2 Report

Corrections are acceptable.

Author Response

Corrections are acceptable.

     We thank the opinion of the reviewer about our work.
